# The Novel Protease Activities of JMJD5–JMJD6–JMJD7 and Arginine Methylation Activities of Arginine Methyltransferases Are Likely Coupled

**DOI:** 10.3390/biom12030347

**Published:** 2022-02-23

**Authors:** Haolin Liu, Pengcheng Wei, Qianqian Zhang, Zhongzhou Chen, Junfeng Liu, Gongyi Zhang

**Affiliations:** 1Department of Immunology and Genomic Medicine, National Jewish Health, Denver, CO 80206, USA; liuh@njhealth.org (H.L.); weip@njhealth.org (P.W.); 2Department of Immunology and Microbiology, School of Medicine, Anschutz Medical Center, University of Colorado, Aurora, CO 80216, USA; 3State Key Laboratory of Agrobiotechnology, Chinese Agriculture University, Beijing 100193, China; b20183020120@cau.edu.cn (Q.Z.); 07006@cau.edu.cn (Z.C.); jliu@cau.edu.cn (J.L.)

**Keywords:** JMJD5, JMJD6, JMJD7, Jumonji, PRMT

## Abstract

The surreptitious discoveries of the protease activities on arginine-methylated targets of a subfamily of Jumonji domain-containing family including JMJD5, JMJD6, and JMJD7 pose several questions regarding their authenticity, function, purpose, and relations with others. At the same time, despite several decades of efforts and massive accumulating data regarding the roles of the arginine methyltransferase family (PRMTs), the exact function of this protein family still remains a mystery, though it seems to play critical roles in transcription regulation, including activation and inactivation of a large group of genes, as well as other biological activities. In this review, we aim to elucidate that the function of JMJD5/6/7 and PRMTs are likely coupled. Besides roles in the regulation of the biogenesis of membrane-less organelles in cells, they are major players in regulating stimulating transcription factors to control the activities of RNA Polymerase II in higher eukaryotes, especially in the animal kingdom. Furthermore, we propose that arginine methylation by PRMTs could be a ubiquitous action marked for destruction after missions by a subfamily of the Jumonji protein family.

## 1. Introduction

Arginine methylation, a ubiquitous post-translation modification (PTM) of proteins, was discovered more than 50 years [1] and is appreciated in recent decades [2,3,4,5,6]. Nine arginine methyltransferases (PRMTs) have been characterized [3]. Among them, PRMT1/2/3/4/6/8, characterized as type I, are responsible for asymmetric methylation on the sidechain of arginine; PRMT5 and PTMT9, as type II, are responsible for symmetric methylation, while PRMT7 is the only member of type III for monomethylation with preferred sites containing sequences of RGG/RG, RXR, GRG, proline-rich, proline–glycine–methionine-rich motifs [3,5]. The exact function of arginine methylation on proteins is still controversial; however, accumulating data about its roles within a large number of RNA-binding proteins with intrinsic disorder regions (IDRs) or low complexity domains (LCDs) showed that it plays critical roles in the phase separation of these proteins and closely relates to neurodegenerative diseases, cancers, and other diseases [7]. Most of these IDR-containing proteins are responsible for the formation of a large number of membrane-less organelles (MLOs) including the nucleus, nuclear speckles, nuclear stress bodies, histone locus body, Cajal body, PML nuclear body, paraspeckles, perinucleolar compartment, stress granules, P-bodies, germ cell granules/nuage, neuronal granules, etc. [8,9]. As we know, most RNAs are vulnerable to attacks by nucleases within cells, and all aforementioned cell bodies could build up safe microenvironments for a variety of protection purposes, including rRNA biogenesis, mRNA splicing, generations of microRNA and snRNAs, mRNA transporting either from the nucleus to cytosol or between germ cells (such as mRNAs from neighboring nutrient cells (germ cell granules) to oocytes) in germ cell development, translation, preservation after stress (stress granules), etc. A large number of research studies showed that arginine methylation within these IDRs is critical for them to participate in the formation of these cell bodies. In this regard, it predicts that the removal methyl groups of these methylated proteins should also be highly involved. However, enzymes that remove these modifications still remain a mystery. The existence of all these bodies is timely; for example, even nucleolus will disassemble during mitosis, not to mention other temporally forming bodies. In this regard, we attempt to speculate that arginine methylation will end up with degradation of the target protein coupling with finished missions. For example, we recently found that JMJD6 could specifically recognize an arginine-methylated arginine cluster to make cleavage at the methylated arginine within the MePCE protein [10], which is a major component of 7SK snRNP complex sequestering p-TEFb (including CDK9 and Cyclin T1) [11]. Cleavage of MePCE by JMJD6 disrupts the inhibitory complex, so as to release p-TEFB, which consequently is recruited to the super elongation complex (SEC) by BRD4 to phosphorylate C-terminal domain (CTD) of RNA polymerase II (Pol II) [10].

On the other hand, histone modification, coupled with transcriptional activities of RNA Polymerase II, is the hallmark of epigenetics. Most modifications including histone lysine methylation, histone ubiquitination, histone phosphorylation, histone acetylation, etc., which are generated by writers [12], are reversible and coupled with enzymes that remove these specific modifications, termed as erasers [13]. The modification of arginine methylation of histone subunits is quite ubiquitous, with multiple methylation sites within each histone subunit [3,14,15]. Compared with lysine methylation, which is very specific and includes H3K4, H3K9, H3K27, H3K36, H3K79, H4K20, etc. with specific roles through recruiting designated functionally defined complexes [16], there is no obvious pattern of arginine methylation, and though a few of them could be recognized by some complexes such as Tudor domain-containing protein, Rag1-PHD, WD4-containing protein, UBR1, etc. [17], most of them are still unaccountable (Figure 1). Furthermore, enzymes responsible for removing these modifications still are not consolidated; two reports showed that JMJD6 could act as an arginine demethylases [18,19], while another report found that JMJD1B may work on methylated H4R3 [20], and two other reports claimed that some lysine demethylases could also work on methylated arginines in vitro [21,22]. We surreptitiously discovered that JMJD5 and JMJD7 could specifically cleave at arginine-methylated sites as endopeptidases and continue to act as amino exopeptidases to trim histone tails [23,24]. These novel discoveries, combined with other data in the transcription field, lead to an unprecedented theory—namely, that the cleavage of arginine-methylated histone tails on +1 nucleosomes at promoters of the stimulation-regulated genes is coupled with the release of paused Pol II in higher eukaryotes [25,26]. Similar to the destiny of arginine-methylated RNA-binding IDR proteins, arginine-methylated histone subunits could be also doomed for destruction. Overall, we speculate that arginine methylation has some similar features as those of ubiquitination, which mostly end up with proteasome-orientated degradation after missions [27].

## 2. Arginine Methylation and Phase Separation for Ribonucleoproteins (RNPs)

A large number of proteins in eukaryotes have IDRs with some repeating low complexity domains containing aromatic rich residues such as tyrosine, as well as arginine-rich repeat domains such as RGG/RG [9,28]. It is very well recognized that these two motifs could make cation–π interaction, which is the driving force for phase separation [29,30,31,32,33,34]. However, it is still controversial whether arginine methylation enhances or suppresses the phase separation. Some reports seem to support the latter [30,31,32,33,35,36,37], though there is a plethora of data supporting that arginine methylation plays a critical role in promoting phase separation [38,39,40,41,42,43,44,45,46,47,48,49]. As regards cation–π interaction, arginine methylation enhances the interaction between the methylated sidechain of arginine and the aromatic sidechain. This concept is very well supported by numerous structural and biochemical data since either lysine methylation or arginine methylation increases the binding affinity between methylated lysine or arginine and the correspondent binding partner. A pioneering structural and biochemical analysis of H3K4me3 and PHD domain from Dr. Dinshaw Patel’s group revealed that the binding affinity increases with methylation of lysine from monomethylation to trimethylation [50]. The rich aromatic side-chain within the PHD domain accounts for the strengthening interactions (Figure 2A). This is also true for the interaction between methylated arginine and Tudor domain from PIWI-binding proteins; an aromatic cage holds the methylated guanidinium moiety, as reported by Drs. Tony Pawson and Jinrong Min’s groups [51], in which methylation increases the binding affinity from ~94 µM to ~10 µM, almost 10-fold (Figure 2B). Interestingly, the catalytic core of JMJD5 also owns a Tudor domain-like structure with a rich aromatic side chain to build a cage to specifically bind to methylated arginine instead of methylated lysine [23,24]. The methylation of arginine of histone H3R2 could enhance the binding affinity from 7 µM to 100 nM, almost ~70-fold (Figure 2C). Interestingly, judging from one report of FUS phase separation, arginine methylation seems to promote membrane-less droplets to form much more tightly ordered spherical shapes; without methylation, however, they have disordered shapes though they become larger [31]. These phenomena were also indicated in the other three reports [30,32,33]. However, it is beyond the scope of this study to interpret other researchers’ data. 

Taken together, it is likely that arginine methylation could promote the property of self-/intra-oligomerization such as in FUS, EWS, TAF15, etc., to form membrane-less organelles. At the same time, arginine methylation generates a docking site for the Tudor domain, which brings two or more molecules together for inter-oligomerization to form isolated particles such as TDRD proteins and SMN proteins [17,52]. Arginine methylation also creates sites for the recognition by other domains. Finally, and most importantly, all these bodies or particles formed through phase separation create relatively isolated microenvironments for RNA splicing, rRNA biogenesis, DNA repair, generation of micro-RNAs, small interfering RNAs, mRNA biogenesis, protection, transporting, etc. 

## 3. Arginine Methylation of Histone Subunits and Transcription Activation

Although arginine methylation on histone also occurs in singular cells, it is mostly related to the heterochromatin regions with repressed transcription activity [14]. However, arginine methylation on histone tails participates in the transcription activation in higher eukaryotes and possibly couples with the Pol II-pausing regulation, a unique transcription regulation mechanism that only occurs in higher eukaryotes. 

It is reported that PRMT1 and CARM1 (PRMT4) function synergistically with each other [53,54]. CARM1 associates with coactivator glucocorticoid receptor-interacting protein 1 (GRIP1) to stimulate transcriptional activation [55]. PRMT1 is recruited by nuclear receptors or coactivators to methylate histone H4R3, while CARM1 works on histone H3 [56,57,58,59]. Knockouts of PRMT1, which are responsible for over 85% of arginine methylation in vivo [60], are embryonic lethal not beyond E6.5 [61,62], while knockout of CARM1 is neonatal lethal [63]. ChIP-seq data show that both CARM1 and its potential methylation target H3R17 are located at the promoter region or transcription start site [64], similar to PRMT2 and its potential substrate H3R8 [65]. PRMT6 deposits H3R2me2a at promoter and enhancer regions [66]. H3R2me2a occupancy at both enhancer and promoter regions drastically increases upon activation by all-trans retinoic acid through a nuclear receptor [66], further confirming earlier reports that PRMTs are recruited by nuclear receptors or their coactivators [56,57,58,59]. It has been a major mystery as to why transcription activation needs arginine methylation on +1 nucleosomes (Figure 3). 

It is very well established that H3.3K4me1 at enhancer regions are generated by MLL3/4 [67,68,69,70,71]. It was found that there is no methylation of H3.3K27 marks at enhancer regions before zygotic genome activation (ZGA) [72,73,74,75]; therefore, MLL3/4 does not need to recruit UTX or KDM7 to generate H3K27me0 sites at this time. However, there are scarce data regarding how MLL3/4 is recruited to enhancer regions. Two reports showed that the ectopic expression of CEBPβ or HOXA9 is sufficient to bring MLL3/4 to enhancer regions, to generate H3K4me1 [67,76], suggesting that MLLL3/4 could be recruited by p300/CBP, coupled with transcription factors. One report shows that the tandem of PHD4-6 of MLL4 is essential for the functioning of MLL4 and that it could specifically bind to H3R3me0 and H4R3me2a but not H4R3me2s [77], suggesting that arginine methylation on +1 nucleosome could be critical for recruiting MLL3/4. It was later confirmed that H4R17 plays an essential role in recruiting MLL3/4 [78]. Interestingly, another report showed that acetylated H4K16ac binds to the PHD6 finger of MLL4 [79]. We propose that both methylated arginines generated by PRMT1/CARM1 on +1 nucleosome and acetylated H4 generated by CBP/p300 on the enhancer nucleosome (-N) could work together with additional elements, such as the enhancer DNA sequence, to bring MLL3/4 to the enhancer region, generating H3.3K4me1 at the enhancer region (Figure 4). Further characterization of the binding specificity of these PHD fingers with MLL3/4 is required.

Interestingly, it was reported that PRMT1 is required to recruit JMJD2C, to trigger transformation in acute myeloid leukemia through removing methyl groups on H3K9 [80]. This is consistent with our recent discoveries that the Tudor domain of the JMJD2 family prefers binding to arginine-methylated histone tails instead of lysine-methylated histone tails (Liu et al., unpublished). It may be known that JMJD2 family members are lysine demethylases that remove methyl groups on H3K9 and H3K36, as characterized by several groups including ours [81,82,83,84]. It was reported by Dr. Bruno Amati’s group that methylation of histone H3R2 by PRMT6 and H3K4 by an MLL complex (which is likely dominant at -1 nucleosome) are mutually exclusive [85], while Dr. Ernesto Guccione’s group revealed that H3K4me2s is rich on +1 nucleosome [86], similar to H4R3me2s found to accumulate at promoter regions [87]. Based on the above-collected data, we currently hypothesize that arginine-methylated histone tails on +1 nucleosome recruit JMJD2 family members to remove methyl groups of H3K9 on enhancer nucleosomes to convert it from inactive enhancer to active enhancer coupled with the function of MLL3/4 (Figure 5). 

In summary, arginine methylation of histone tails on +1 nucleosomes generates docking sites for the JMJD2 family (through Tudor domain), MLL3/4 (through plant homeodomain, PHD, and/or WDR domain-containing proteins within COMPASS-like complex), etc. (Figure 3, Figure 4 and Figure 5). JMJD2 family members, joined by other monomethyl-removing Jumonji family members, remove methyl groups on H3K9 (or H3K36 if the enhancers are located within intron regions) of nucleosomes at enhancer regions (-L, -M, -N), which are further modified by MLL3/4 to generate H3K4me1 on nucleosomes at enhancer regions. Arginine methylation on +1 nucleosomes of genes controlled by stimulating signals could be an essential step for transcription activation of genes regulated by enhancers, and it is likely coupled with Pol II pausing. 

## 4. Arginine Methylation and Transcription Repression

Compared with PRMT1- and CARM1 (PRMT4)-accompanying transcription activation [53,56,58,88,89], PRMT5 and PRMT6 are always coupled with the transcription repression process (Figure 6). PRMT5 symmetrically di-methylates H2AR3, H4R3 [90], and H3R8 to mediate transcriptional repression [91,92]. PRMT6 functions mainly as a transcription co-repressor by asymmetrically di-methylating H3R2 [85] or H2AR29 [93]. It remains unknown why transcription repression requires arginine methylation on histone tails. Interestingly, Tudor domains could be found in a series of proteins, which participate in transcription repression or heterochromatin formation. An H3K9 methyltransferase SETDB1 contains two Tudor domains [94,95]. It is reported that a subset of the histone H3K9 methyltransferases Suv39h1, G9a, GLP, and SETB1 form a complex to function together [96]. It is obvious that methylated arginines of histone tails on nucleosome +1 may recruit SETDB1 to convert H3.3K9me0 to H3.3K9me3 on nucleosomes at the enhancer region (-N) with the help of other H3K9 methyltransferases, which can be specifically recognized by the heterochromatin protein 1 (HP1/CBX5) family [97], to form constitutive heterochromatin for a complete silence of a target gene (Figure 7 and Figure 8). A large number of PRC2-associated protein families, PFH, also contain Tudor domains [98,99]. It is likely that they may help PRC2 to be recruited to promoter regions to shut down the transcription unit. Interestingly, Tudor-domain-containing proteins ARID4A and ARID4B [98] are found to associate with Sin3/Rpd3 repression complex [100], which is a major histone deacetylase complex to remove acetyl groups on histone tails. 

In summary, several Tudor-domain-containing transcription cofactors that are involved in transcription repression could be recruited by arginine-methylated histone tails on +1 nucleosomes of stimulation-regulated genes, with correspondent repressing complexes such as Sin3-containing histone deacetylase complex, which remove acetyl groups on nucleosomes at enhancers, or PRC2 complex, which builds facultative heterochromatin to convert an active transcription unit back to a repressive unit for a later reactivation, a very popular action after zygote activations during embryonic development. However, data on direct and specific interactions between these Tudor domains of cofactors and arginine-methylated histone tails are still lacking. This could be a future exciting field to explore. 

## 5. The Ubiquitous Arginine Methylation and Potential Final Destination

Proteomic analysis of the status of arginine methylation reveals an astonishing fact—namely, that arginine methylation is similar to those of phosphorylation and ubiquitination; it is ubiquitous (Figure 1) [15]. It is very well established that both phosphorylation and ubiquitination are reversible, which are achieved by either large family members of phosphatases [101] or a large group of deubiquitinating enzymes [102]. However, the hunting of arginine demethylases is not very successful so far. It is of interest to know whether it exists in vivo and, most importantly, the exact biological consequence of the actions. As mentioned above, arginine methylation is not simply required for some stimulating signals such as heat shock, DNA repairing, differentiation cues, etc., but also for phase separation of membrane-less bodies to serve different purposes in RNAs, such as splicing, rRNA biogenesis, snRNAs and microRNAs biogenesis, etc. for non-histone proteins. In this regard, we tend to speculate one destiny of arginine methylation: Irrespective of the purpose they serve in either non-histone proteins or histone subunits, they are doomed for destruction after missions accomplished, similar to a popular ubiquitination pathway—the proteasome-orientated degradation. The most outstanding feature for targets bound to ubiquitination degradation is frequently to be regenerated without major economic burdens. As we know, most of the cell bodies are disassembled; even nucleolus is dissolved during mitosis. On the other hand, accumulating data suggest that arginine methylation on +1 nucleosomes of stimulating genes also fits this category; a gained property during evolution in higher eukaryotes, especially in the animal kingdom, it may be born to be destroyed to regulate transcription activities of a unique group of stimulating genes. The discoveries of proteases activities of JMJD5, JMJD6, and JMJD7 from our group indicate there does exist a novel destruction mechanism of arginine-methylated proteins, as described below [10,23,24,25,26,103].

## 6. The Novel Protease Activities of JMJD5 Arginine-Methylated Histone Tails Coupled with CDK9 to Release Paused Pol II

After some pioneering studies in characterizing lysine demethylases of the Jumonji protein-containing JMJD2 subfamily [81,82,83], we sought to identify potential candidates of arginine demethylases from this same Jumonji protein family. The first candidate we attempted to identify was JMJD6. However, we failed to detect any activities of JMJD6 toward methyl groups on arginines of histone tails, as we mention in the next section. At the same time, there was also some controversy regarding the function of JMJD5 [104], which was first characterized as lysine demethylases [105], while another group found it has an arginine hydroxylase activity [106]. JMJD5 seems to play a critical role in the early development of mice since knockout leads to early embryonic lethality [107,108,109]. Interestingly, we detected a drop in the content of methylated arginines when bulk histone was treated with JMJD5 [23]. However, it was impossible for us to identify any activities of removal of methyl groups on potentially methylated arginines of histone tails with synthesized peptides. To avoid the nonspecific issues of antibodies used for the readout of methylated arginines, we generated C^14^ radioactive-methylated histone tails by treating bulk histone with PRMT1/5/6/7 and C^14^-SAM [23]. These C^14^-positive substrates were subjected to an enzymatic reaction of JMJD5. To our surprise, short fragments started to appear after treatment of JMJD5 [23]. Follow-up characterization revealed that JMJD5 owns both endopeptidase and carboxy–exopeptidase activities, a novel protease family (Figure 9) [23], which contains all essential structural features to act as a hydrolase (Figure 10). Interestingly, another group also found protease activities of JMJD5 on H3, though specific for lysine residue later [110]. Furthermore, we found that there is a unique substrate recognition feature within the Tudor-domain-like motif, which could discriminate the side chain of arginine from that of lysine, suggesting a very specific recognition mode between JMJD5 and methylated arginines (Figure 10) [24]. Another surprising discovery is that JMJD5 affects the homeostasis of both arginine-methylated histones and histone overall; depletion of JMJD5 leads to the dramatic accumulation of both components in MEF cells or human cancer cells [23], providing strong evidence to support the cleavage roles of JMJD5 on arginine-methylated histones. These data suggest that arginine-methylated histone on +1 nucleosomes, as we discussed earlier, are doomed for destruction instead of recycling through demethylation or reversible recovery, a hallmark of epigenetics. Our later ATAC-seq data showed that the location of +1 nucleosomes from a large number of genes shifted upstream in the JMJD5 knockout MEF cells, suggesting JMJD5 only works on +1 nucleosomes of some stimulating genes [25]. However, this novel discovery also raises a significant question on how JMJD5 is recruited and carries out its function. This question actually leads to another novel discovery, which may raise the curtain on the unique mysterious transcription regulation mechanism of promoter-proximal Pol II pause in higher eukaryotes.

There was an uncharacterized N-terminal domain (NTD-JMJD5) with a flexible linker to connect to the C-terminal catalytic core domain of JMJD5. From secondary and three-dimensional structural predictions, we found that this N-terminal domain is quite similar to those of NRD1, PCF11, Ritt103, SCAF8, and RPRD1A/B, well-recognized C-terminal of Pol II (CTD-Pol II) binding proteins [111,112,113,114,115]. This observation guided us to explore the potential association between NTD-JMJD5 and phosphorylated CTD-Pol II. We found that NTD-JMJD5 could pull down a very special specie of phosphorylated CTD-Pol II [25], which is only recognized by a rabbit polyclonal antibody generated from CTD-heptad repeats with phosphorylated serine-2 within each repeat (-YS(p)PTSPSYS(p)PTSPS-) [116] but not by a widely used monoclonal antibody 3E10, which was raised using a single phosphorylated Serine-2 CTD-heptad peptide (-YS(p)PTSPS-) [117]. Further characterization revealed that NTD-JMJD5 has a very high binding affinity toward a CTD-heptad repeating peptide with both serine-2 phosphorylation and additional Serine-5 phosphorylation in the second repeat (-YS(p)PTSPSYS(p)PTS(p)PS-) with an extremely high binding affinity (~9 nM) [25]. Interestingly, this unique phosphorylation pattern of CTD-Pol II actually has been revealed in higher eukaryotes but not in yeast, though the authors and reviewers may have neglected novel differences and significances [118]. This novel discovery leads to another significant question: Which kinase is responsible for the generation of this unique phosphorylated CTD-Pol II pattern in vivo? As reported early, CDK9 could phosphorylate the serine-2 of CTD-Pol II at the early stage of transcription and is critical for the release of paused Pol II in higher eukaryotes [118,119,120,121,122,123,124,125], which is unique in higher eukaryotes, though some reports claimed that Bur1 is the homolog of CDK9 in yeast [126,127], which is still a hotly debated topic in the transcription field. In line with our expectations, inhibition by flavopiridol or depletion through ubiquitin targeting of CDK9 lead to the dramatic drop in this phosphorylation pattern of CTD-Pol II [25]. Based on (1) the early discoveries, which showed that +1 nucleosome is the cause of Pol II pause [128,129,130,131], (2) our findings that JMJD5 cleaves specifically on arginine-methylated histone tails on +1 nucleosomes to generate “tailless nucleosomes” [23,25], and (3) the recruitment of JMJD5 by the phosphorylated CTD-Pol II generated by CDK9 [25], we concluded that JMJD5 might couple with CDK9 to release paused Pol II in higher eukaryotes for the stimulation-regulated genes [25].

## 7. JMJD6 Cleaves MePCE to Disrupt the 7SK snRNP Complex to Release p-TEFb

JMJD6 is one of the most controversial proteins in the field of biology [132]. It was first cloned as phosphatidylserine (PS) receptor [133] but was corrected as a nucleus-existing protein unrelated to PS [134,135,136]. It was later reported to contain arginine demethylase activity on histone tails [18], hydroxylase activity on splicing factor U2AF65 [137] and histone tails [138], and both arginine demethylase activities on histone tails and RNA demethylase activities on 5′ prime of 7SK snRNA [19], and, surprisingly, PS binding [139,140]. The exact or cognate substrate(s) of JMJD6 remained elusive, though it was found that JMJD6 belongs to a functionally not very well characterized Jumonji domain-containing hydroxylase family [134,135,136]. After determining the structure, we found that JMJD6 has some unique structural features besides similarities to a hydroxylase family member known as the factor of hypoxia-inducing factor 1 inhibitor (FIH-1) [141]. The structural information guided us to characterize the function of the JMJD2 family, one of the pioneering members of which has been identified as histone lysine demethylase [81,82,83]. However, we had a hard time finding any enzymatic activities of JMJD6 toward either methylated lysine or arginine of histone, which held us back to publish it, until several years later, when we found that it nonspecifically recognizes single-strand RNA (ssRNA) through its disordered C-terminal arginine-rich motif with high affinity (~40 nM) [103]. We speculated that it could be an RNA demethylase based on its tight binding to the ssRNA [103]; this assumption proved incorrect, based also on our current discoveries [10]. Interestingly, we indeed observed a loss in arginine-methylated histone tails using bulk histone as substrate by specific antibodies against arginine-methylated histone tails when JMJD6 was added [23]. This may explain why JMJD6 was identified as a histone arginine demethylase [18,19].

Several lines of evidence drove us to explore the protease activities of JMJD6. First, a noteworthy report from Dr. Michael Rosenfield’s group showed that JMJD6 could destroy the 7SK snRNP complex to release P-TEFb [19]. Second, the discoveries of the unexpected proteases activities of JMJD5 and JMJD7 on histone tails during the characterization of the protease activities of JMJD5 and JMJD7 attest that it is possible that JMJD6 could also act as a protease [23]. Third, we indeed found unspecific protease activities of JMJD6 when it was applied to bulk histones [23]. Finally, it is straightforward for us to select 7SK snRNP as a potential substrate based on the findings of Dr. Rosenfield’s group after we failed to identify specific activities of JMJD6 on histone tails. Based on the novel protease activities of JMJD5 and JMJD7 [23,24], the high structural similarity among catalytic cores of JMJD5, JMJD6, and JMJD7 [24,103], and severe phenotypes among knockouts of JMJD6 and JMJD5 in mice [107,108,136,142], we hypothesized that JMJD6 may contain protease activity working on methylated arginines on some protein candidates, which regulate the activity of Pol II, especially promoter-proximal paused Pol II. This was found to be true, as JMJD6 specifically cleaves an arginine-rich sequence (-KRRRR-) site within MePCE, a major component of the 7SK snRNP complex (Figure 11) [10], which primarily functions to sequester the CDK9-containing p-TEFb [143,144]. Methyl phosphate capping enzyme (MePCE) was first characterized as a component of the 7SK snRNP complex that acts as a capping enzyme on the gamma phosphate at the 5′ end of 7SK RNA [145], though another group claimed it also has RNA methyltransferase activities on 5′ phosphate of microRNAs [146]. Furthermore, a capping-independent function of MePCE via stabilization of 7SK snRNA and facilitation in the assembly of 7SK snRNP was reported by Dr. Qiang Zhou’s group [147]. Knockdown of MePCE led to the destabilization of the 7SK snRNP complex in vivo [147,148,149]. We found that depletion of MePCE dramatically increased the activities of CDK9, which is consistent with the discoveries reported previously [147,148,149]. Most importantly, the novel protease activity of JMJD6 toward MePCE elucidates the underlying mechanism of how the activity of CDK9 is strictly controlled and requires the help of both BRD4 and JMJD6, further suggesting that there is virtually no free CDK9 complexes for super elongation complex (SEC) to recruit under normal circumstances. Based on these discoveries, we proposed that JMJD6 cleaves MePCE to release p-TEFb [10].

On the other hand, it appears that the super elongation complex (SEC) is unlikely to recruit P-TEFb without the assistance of JMJD6 and BRD4. Compared with efficient recruitment of p-TEFb by TAT protein in the human immunodeficiency virus (HIV) [150], BRD4 is claimed to be responsible for the endogenous recruitment of p-TEFb to the promoters of Pol II pause-regulated genes [143,144,149]. However, BRD4 lacks an RNA-binding motif, compared with TAT (Wu et al. 2007). Therefore, we hypothesized that there must exist another factor to help BRD4 recruit p-TEFb and engage in the instigation of Pol II transcription elongation. Besides the classic Bromo domains, which recognize acetylated histone tails, BRD4 contains an extra terminal domain (ET) that recognizes JMJD6 [151,152]. Combined with our discoveries that JMJD6 nonspecifically binds to single-stranded RNA with high affinity [103], these findings led us to propose that JMJD6 may be recruited by both BRD4 and newly transcribed RNAs from Pol II, to help BRD4 recruit p-TEFb, acting analogously to that of TAT protein associating with both p-TEFb and TAR [10].

Interestingly, JMJD6 was found to coexist with stress granule (SG)-related protein G3BP1, and removal of JMJD6 leads to the accumulation of arginine-methylated G3BP1 and affect its function of SG formation; therefore, the authors further suggested that JMJD6 could be an arginine demethylase [37]. It will be of interest to investigate whether JMJD6 cleaves arginine-methylated G3BP1 and affect the overall homeostasis of G3BP1. If this is true, the above result could also be interpreted alternatively—namely, that JMJD6 may cleave arginine-methylated G3BP1 and lead to final degradation of G3BP1, so as to resolve SGs when the challenge is eliminated.

## 8. The Protease Activity of JMJD7 on Histone Tails and Beyond

JMJD7 is a barely touched Jumonji protein family member. Based on the sequence similarity with JMJD5 at the catalytic core, the enzymatic activities of protease on arginine-methylated histone tails were characterized simultaneously with JMJD5 by us (Figure 9) [23]. Further structural and functional characterizations revealed that there is a high structural similarity between JMJD7 and JMJD5, with specificity toward arginine-methylated histone tails but different from JMJD5 such as different sensitivity toward combinations of modification of histone tails [24]. Knockout of JMJD7 in a human cancer cell line dramatically represses the growth of the cell, suggesting a critical role in cell proliferation [23]. Furthermore, the depletion of JMJD7 also leads to the accumulation of the contents of arginine-methylated histone, as well as the overall histone, suggesting its similar role as that of JMJD5 in regulating the homeostasis of the histone [23]. Interestingly, a recent proteomic analysis showed that JMJD7 is associated with several transcription factors, such as FOXI1 and Pogo transposable element with ZNF domain (POGZ) [153], the latter of which is a critical transcription factor to regulate human fetal hemoglobin expression [154], as well as having critical roles in neuron development in the brain [155]. An early report showed that JMJD7 and POGZ work together to regulate the differentiation of Osteoclast, while JMJD7 was found to occupy the promoter regions of several genes associated with the Osteoclast differentiation [156], suggesting that JMJD7 is recruited to the promoter regions of these genes and possibly required for activation of these genes. Another important feature of JMJD7 is that it has been frequently found to fuse with a phosphatase, PLA2G4B, and regulates the proliferation of the head and neck squamous cell carcinoma [157], though the underlying mechanism remains to be investigated. Interestingly, another group claimed that JMJD7 is a lysine hydroxylase, specific for two translation factors—TRAFAC and GTPases [158]; it will be of interest to find the consequence of hydroxylation of these GTPases.

Compared with JMJD5, JMJD7 lacks a similar N-terminal domain, which is required for the recruitment of JMJD5 to Pol II. It is likely that JMJD7 may be recruited through another mechanism, such as direct transcription factor recruitment as POGZ to the promoter regions of regulated genes to cleave arginine-methylated histone tails and lead to the similar “Tailless nucleosomes” as that of JMJD5. There are numerous questions that remain to be answered. Most importantly, there is a lack of a knockout model of JMJD7 in mice. Proteomic analysis of JMJD7-associated partners is also an exciting direction to explore.

## 9. Cancers Coupled with Upregulations of JMJD5/JMJD6/JMJD7 and PRMTs

Based on the important roles of JMJD5/6/7 in the development of embryos and transcription activations in higher eukaryotes, as analyzed above, it is not surprising that all of them are upregulated in various cancers. JMJD5 is highly expressed in breast cancer [105], lung cancer [159], colon cancer [160], prostate cancer [161], etc. JMJD6 is also found to express highly in numerous cancers [162,163,164,165,166,167,168,169]. Even JMJD7 fused with PLA2G4B is elevated in head and neck squamous cells [157]. In this regard, inhibitors of JMJD5, JMJD6, and JMJD7 could be effective anticancer drug targets to treat different cancers. Interestingly, inhibitors of JMJD6 have been developed and have shown repressive effects on some types of cancers [163,170,171,172,173,174,175]. There is no report about inhibitors on JMJD5 and JMJD7 yet. However, following the same facts, these inhibitors could be toxic to animals, including human beings, due to the critical function of JMJD5, JMJD6, and JMJD7. PRMTs are highly involved in cancers, and inhibitors have been developed to treat cancers. It is currently a highly active topic in the field of anticancer drug development [2,5].

## 10. Conclusions and Perspectives

Based on the detailed analysis presented in this study of the novel protease activities of JMJD5/6/7 and ubiquitous activities of PRMTs, we speculate that these two enzyme systems may build up a novel protein destruction pathway similar to that of ubiquitous pathways but responsible for more specific regulations by participating in the regulation of individual pathways. There are more than 60 members of the Jumonji protein family, and besides JMJD5/6/7, the exact biological function of several members of the small size subfamily including JMJD4, JMJD8, and others are not very well characterized [109]. It will be of great interest to investigate the phenotypes of knockouts of each member. We expect that some of them could act as proteases to digest protein substrates containing methylated arginines. Similar to ubiquitous pathways, it will take considerable effort to build up methodologies to tackle their details. Some significant questions could be addressed, for example, why this pathway instead of the ubiquitous pathway to destroy proteins, how the two pathways are communicating with each other, at what stage they merge together, etc. Interestingly, this report reveals that JMJD5 promotes the ubiquitous-orientated degradation of circadian oscillator protein CRY1 [176]. It will be an exciting topic to investigate the molecular basis of this process.

## Figures and Tables

**Figure 1 biomolecules-12-00347-f001:**
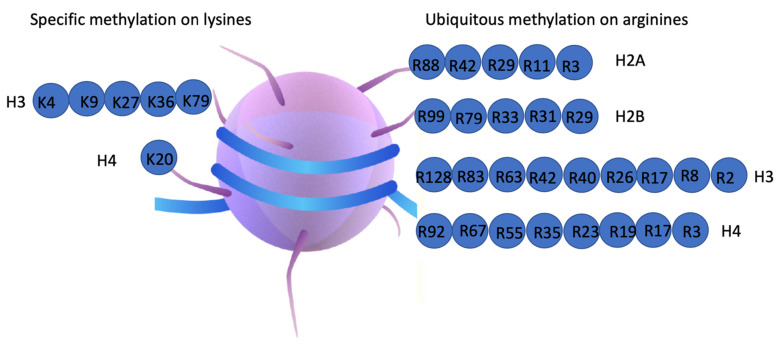
Possible histone methylation. Histone lysine methylation is very specific and limited while histone arginine methylation seems not specific and ubiquitous.

**Figure 2 biomolecules-12-00347-f002:**
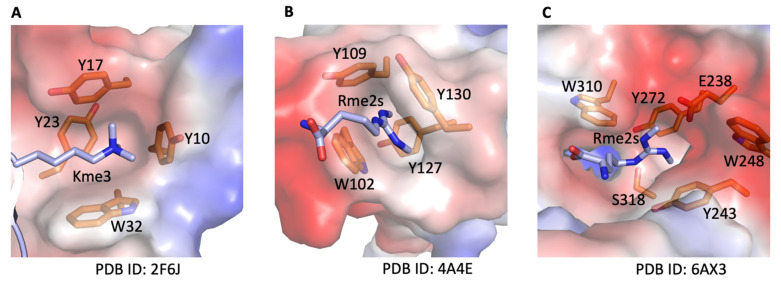
Methylation promotes the interaction between positively charged sidechains (cations) and aromatic cage (π). Kme3, trimethylation of Lysine; Rme2s, symmetric di-methylation of arginine. (**A**). Detailed interactions between PHD domain and tri-methylated lysine. (**B**). Detailed interactions between Tudor domain and symmetrically di-methylated arginine. (**C**). Detailed interactions between the substrate-binding domain of JMJD5 and symmetrically di-methylated arginine.

**Figure 3 biomolecules-12-00347-f003:**
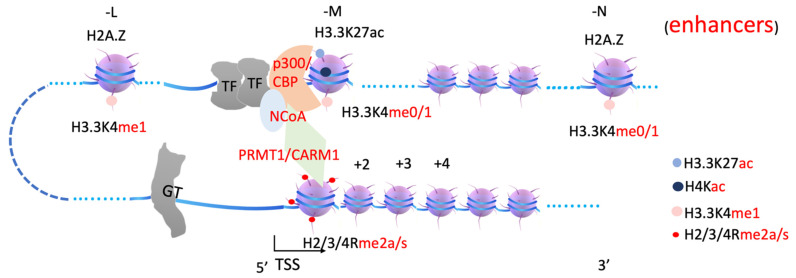
A model of transcription activation and arginine methylation on +1 nucleosome. A transcription unit includes enhancers, transcription factors (TFs), p300/CBP, nuclear receptor co-activator (NCoA), arginine methyltransferase 1 or 4 (PRMT1, CARM1), arginine methylated histone tails at +1 nucleosome. H3.3K27ac, acetylated H3.3 subunit; H4Kac, acetylated H4 subunit; H3.3K4me1, monomethylated H3.3K4; H2/3/4Rme2a/s, arginine methylated histone subunits H2, H3, and H4. GT, general transcription factors. The following figures have the same labels.

**Figure 4 biomolecules-12-00347-f004:**
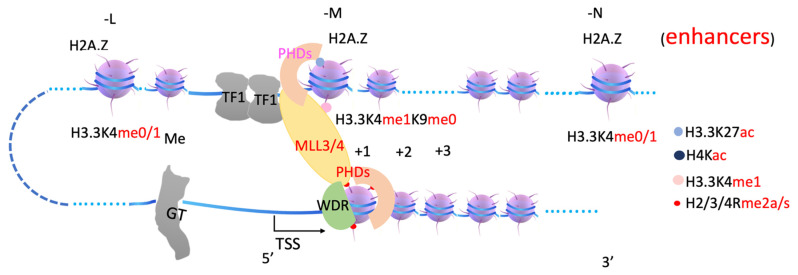
A model of how MLL3/4 is recruited by arginine methylated histones on +1 nucleosome. MLL3/4 complex is recruited by arginine methylated histone tails at +1 nucleosome through PHD domains and/or WDR domain-containing proteins, is responsible for generating H3.3K4me1 at enhancers. MLL3/4, Myeloid/Lymphoid or Mixed-Lineage Leukemia Protein 3/4. WDR, WD repeating-containing protein.

**Figure 5 biomolecules-12-00347-f005:**
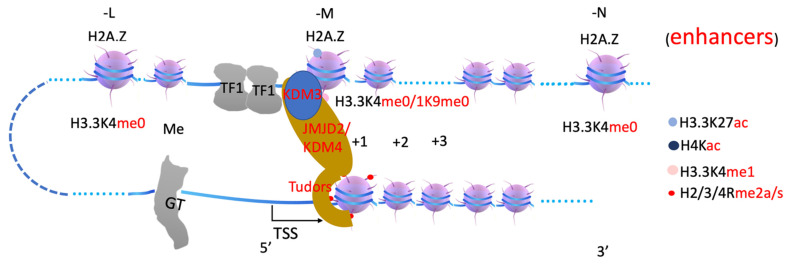
A model of transcription of how KDM3/4 is recruited by arginine methylated histone on +1 nucleosome. KDM3/4 are H3K9 and H3K36 specific lysine demethylases to remove methyl groups on H3.3K9 at enhancer regions. Tudor domain is specific to recognize arginine methylated histone tails at +1 nucleosome.

**Figure 6 biomolecules-12-00347-f006:**
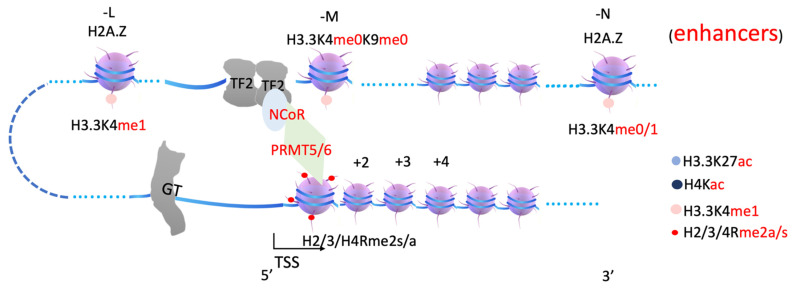
A model of transcription repression and arginine methylation on +1 nucleosome. When transcription repressors (TF2) bind to the enhancer regions, which recruit nuclear receptor co-repressor (NCoR). NCoR, in turn, recruits PRMT5/6 to generate methylated arginine on the +1 nucleosome.

**Figure 7 biomolecules-12-00347-f007:**
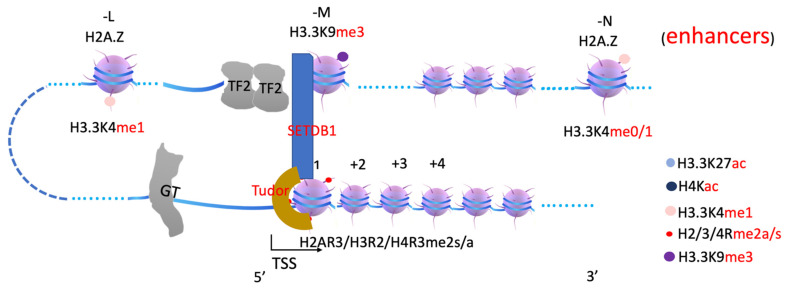
A model of how SETDB1 is recruited by arginine methylated histone on +1 nucleosome. SETDB1 will be recruited through the Tudor domain within SETDB1 and methylates H3.3K9 to generate H3.3K9me3. H3.3K9me3, trimethylated H3.3K9.

**Figure 8 biomolecules-12-00347-f008:**
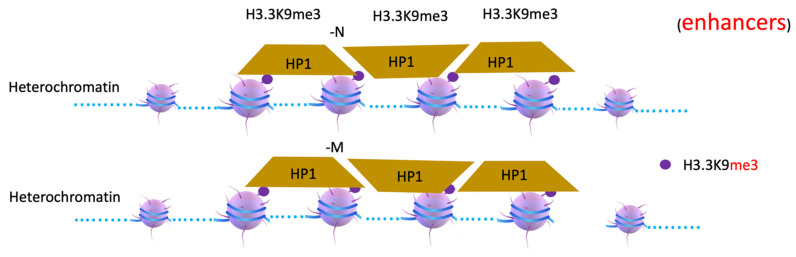
A model of how heterochromatin is formed by HP1 after methylation of H3K9. H3.3K9me3 will recruit HP1protein to form heterochromatin. HP1, heterochromatin protein 1.

**Figure 9 biomolecules-12-00347-f009:**
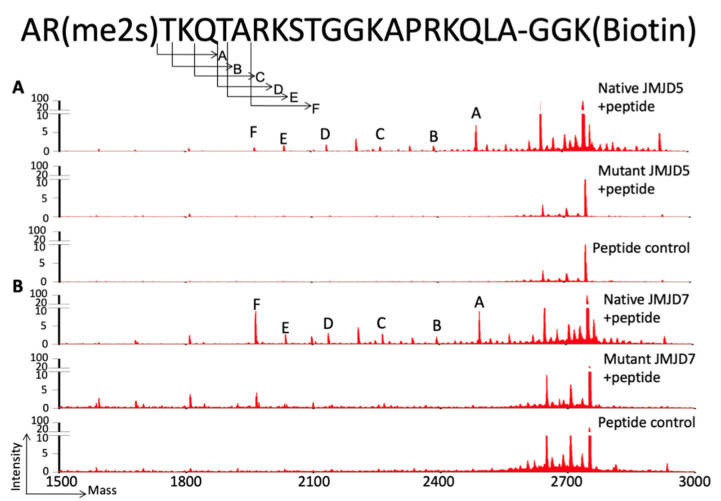
The endopeptidase and carboxy-exopeptidase activities of c-JMJD5 and c-JMJD7 on synthesized histone tails. (**A**). c-JMJD5 and mutant c-JMJD5 cleavage on H3R2me2s. The top portion is the sequence of the H3R2me2s peptide with symmetric di-methylation on R2 with MW 2749.47 Da. After cleavage, a major band of MW 2494.3 (peak A) is the product of peptide with the first two residues missing. (**B**). c-JMJD7 generated a similar profile [23].

**Figure 10 biomolecules-12-00347-f010:**
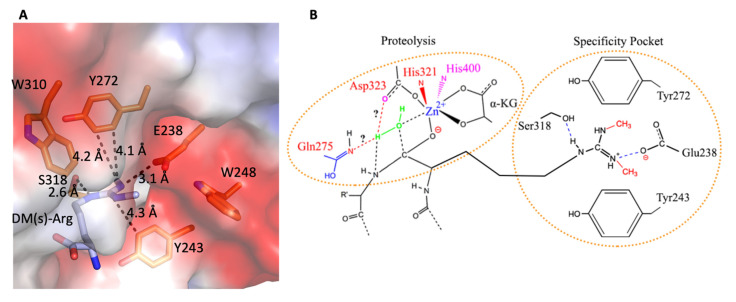
The structural basis of specific recognition between methylated arginine and JMJD5. (**A**). Detailed interactions between demethylated arginine and JMJD5. (**B**). A diagram shows the potential catalysis mechanism and specific recognition between methylated arginine and JMJD5 [24].

**Figure 11 biomolecules-12-00347-f011:**
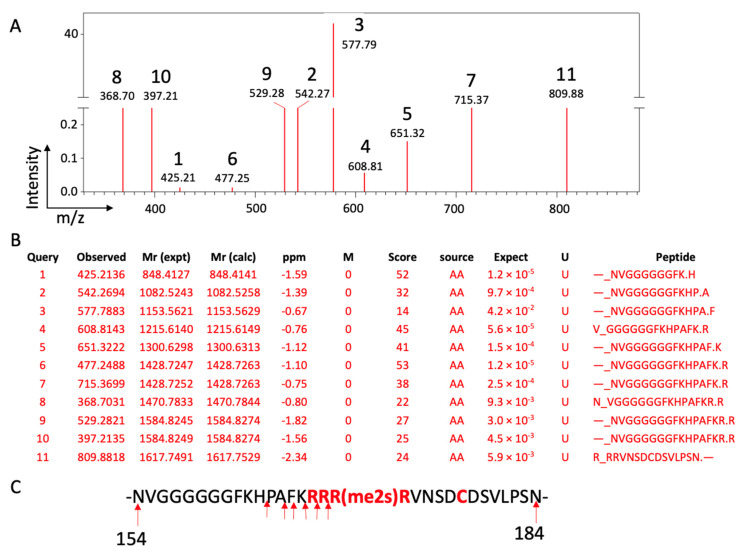
The endopeptidase and amino-exopeptidase activities of JMJD6 on MePCE. (**A**). Mass data from Orbitrap Velos, (**B**). Mass assignments, (**C**). Peptide AA Seq.

## Data Availability

Not applicable.

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
