# Peer review of "The Novel Protease Activities of JMJD5–JMJD6–JMJD7 and Arginine Methylation Activities of Arginine Methyltransferases Are Likely Coupled"

_biomolecules, 2022, doi:10.3390/biom12030347_

Round 1

Reviewer 1 Report

The review entitled 'The novel protease activities of JMJD5-JMJD6-JMJD7 and arginine methylation activities of arginine methyltransferases are 
likely coupled' is a good piece of work. However, the authors need to carry out substantial English language corrections, which could improve the readability of the manuscript. Otherwise, this is a scientifically sound manuscript. 

The manuscript authored by Liu et al. provides evidence that the protease activities of Jumonji proteins, jMjD5/6/7, and methyltransferase activities of arginine methylation are likely linked, contributing to the protein destruction pathway. The authors provide evidence for this based on the studies conducted by them as well as others. This is a good piece of work; the arguments that the authors provided are scientifically sound. However, I found it very difficult to read the manuscript due to grammar mistakes throughout, mixed-use of different tenses, and incorrect sentence structures. I cannot provide examples because there are too many. Therefore, I suggest thorough English language editing to improve the readability. Also, please note that none of the Figures (except Figures 1 and 5) have legends though there is an explanation provided in the text. The figures should be stand-alone, and therefore, legends should be provided.

Author Response

Answer: Thanks for your kind words, we feel sorry for not paying attention to editing.

1)We have edited it carefully in this revision, hope it improves now.

2) We have added details legends on all figures.

Reviewer 2 Report

In this review, the authors are trying to elucidate that the function of JMJD5/6/7 and PRMTs are likely coupled. The review provides a detailed explanation regarding the mechanism of Arginine methylation in transcription activation and repression. Furthermore, the present review is trying to explore the novel protease activities of Jumonji domain containing family including JMJD5, JMJD6, and JMJD7 and their possible functional association with PRMTs. However, the authors should address the following concerns:

  1. Even though the concept of the review is interesting, in order to make the idea more clear, it is suggested to include a table, which summarizes the key events that are coupling the actions of PRMTs and major JMJDs
  2. It is suggested to include a schematic representation for the action and cleavage sites of JMJD 6 & 7.
  3. It is mentioned that JMJD5/6/7 are up-regulated in various cancers. Is there any similar reports, which explains the activities of PRMTs in these conditions? If so, include that as well to make the concept more vivid.
  4. Is there any available literature showing the targeting of JMJD5/6/7 in cancer? If so, include a short summary of that as well.
  5. There are grammatical errors at multiple instances.

The authors should consider addressing these above concerns and improvising the research to make it more impactful. The manuscript requires minor revision

Author Response

In this review, the authors are trying to elucidate that the function of JMJD5/6/7 and PRMTs are likely coupled. The review provides a detailed explanation regarding the mechanism of Arginine methylation in transcription activation and repression. Furthermore, the present review is trying to explore the novel protease activities of Jumonji domain containing family including JMJD5, JMJD6, and JMJD7 and their possible functional association with PRMTs. However, the authors should address the following concerns:

  1. Even though the concept of the review is interesting, in order to make the idea more clear, it is suggested to include a table, which summarizes the key events that are coupling the actions of PRMTs and major JMJDs

Answer: Thanks for your kind words. We are not in the right position to evaluate the field of PRMTs in the aspect of major breakthroughs. People in the epigenetic field still do not appreciate the novel protease activities of JMJDs yet, it will irritate people more if we claimed that our discoveries of the novel function of JMJD5/6/7 as one of the major events in the field. In this regard, we hope other researchers instead of ourselves claim it.

  1. It is suggested to include a schematic representation for the action and cleavage sites of JMJD 6 & 7.

Answer: We add Fig. 9 to show the endopeptidase and carboxy-exopeptidase activities of JMJD5 and JMJD7 on arginine methylated H3R2. We add Fig.11 to show the endopeptidase and amino-exopeptidase activities of JMJD6 on MePCE.

  1. It is mentioned that JMJD5/6/7 are up-regulated in various cancers. Is there any similar reports, which explain the activities of PRMTs in these conditions? If so, include that as well to make the concept more vivid.

Answer: Yes, actually, PRMTs have been reported up-regulated in numerous cancers, people have developed inhibitors to treat different cancers, some of them are in clinical trials. There are lots of existing reviews on this topic, we will mention some of them in the revision. However, there seems no report of coupling of JMJD5/6/7 and PRMTs.

  1. Is there any available literature showing the targeting of JMJD5/6/7 in cancer? If so, include a short summary of that as well.

Answer: Yes, several reports showed that inhibitors of JMJD6 have anti-tumor properties, we added them in context. There is no report of inhibitors on JMJD5/7 yet.

  1. There are grammatical errors at multiple instances.

Answer: Thanks for pointing them out, we did some careful corrections in this version. Hope it improves now.

The authors should consider addressing these above concerns and improvising the research to make it more impactful. The manuscript requires minor revision

Answer: we hope to have addressed the questions raised in the revised version.